# Rainfall as a driver for near-surface turbulence and air-water gas exchange in freshwater aquatic systems

**Eliana Bohórquez-Bedoya**[1,2], **Lorenzo Rovelli**[2,3], **Andreas Lorke**[2]*

**1** Department of Geosciences and Environment, Universidad Nacional de Colombia, Medellín, Antioquia, Colombia, **2** Institute for Environmental Sciences, University of Kaiserslautern-Landau, Landau, Rhineland-Palatinate, Germany, **3** now at the Department of Ecology, Federal Institute of Hydrology - BfG, Koblenz, Germany

* a.lorke@rptu.de

**Data Availability Statement:** Datasets for this research are available in these data citation references: Bohórquez, Eliana; Lorke, Andreas; Rovelli, Lorenzo (2021): Dataset: "Rainfall as a

## Abstract

Gas fluxes from aquatic ecosystems are a significant component of the carbon cycle. Gas exchange across the air-water interface is regulated by near-surface turbulence and can be controlled by different atmospheric forcing conditions, with wind speed and surface buoyancy flux being the most recognized drivers in empirical studies and modeling approaches. The effect of rainfall on near-surface turbulence has rarely been studied and a consistent relationship between rain rate and near-surface turbulence has not yet been established. In this study, we addressed some limitations still present in the quantitative understanding of the effect of rain rate on near-surface turbulence and on the resulting gas transfer velocity in freshwater. We performed controlled laboratory experiments over a wide range of rain rates (7 to 90 mm h$^{-1}$) and estimated gas transfer velocities from high-resolution measurements of $O_2$ concentration, while rain-induced turbulence was characterized based on particle image velocimetry. We found that the rain-induced dissipation rates of turbulent kinetic energy declined with depth following a consistent power-law relationship. Both energy dissipation rates and gas transfer velocity increased systematically with the rain rate. The results confirm a causal relationship between rainfall, turbulence, and gas exchange. We propose a power-law relationship between near-surface turbulent dissipation rates and rain rate. In combination with surface renewal theory, we derived a direct relationship between gas transfer velocity and rain rate, which can be used to assess the importance of short-term drivers, such as rain events, on gas dynamics and biogeochemical cycling in aquatic ecosystems.

## Introduction

Rainfall plays a fundamental role in the biosphere, e.g., in climate, hydrological, and biogeochemical cycles [1, 2]. On water surfaces, rainfall also causes physical impacts [3] that affect, among other processes, the gas exchange across the air-water interface in marine [4] and

Driver for Near-Surface Turbulence and Air-Water Gas Exchange in Aquatic Systems". figshare. Dataset. https://doi.org/10.6084/m9.figshare. 17693792.v1, with the license CC BY 4.0.

**Funding:** This study was partially supported by the German Research Foundation (DFG), project number LO 1150/12-1 (AL) and RO 5921/1-1 (LR). https://www.dfg.de/en/ EBB. was funded by the following programs: - Research Grants - Short-Term Grants, 2019 (57440917) of the German Academic Exchange Service (DAAD), https://www.daad.co/es - Scholarship Program No. 757 - National Doctorates of the Ministry of Science, Technology and Innovation of Colombia, https://minciencias.gov.co/ - the Call for Teaching and Student Mobility of 2019-2021 of the Facultad de Minas of the Universidad Nacional de Colombia, https://minas.medellin.unal.edu.co/ The funders had no role in the study design, data collection and analysis, decision to publish or preparation of the manuscript.

**Competing interests:** The authors have declared that no competing interests exist.

inland waters [5, 6]. The air-water gas exchange has important implications for aquatic ecosystems and global biogeochemical cycles of climate-relevant gases [7, 8], e.g., in regulating the $CO_2$ uptake of the ocean and the greenhouse gas emissions from inland waters [9, 10]. Currently, the prediction of gas exchange rates and their dependence on variable and dynamic environmental conditions are among the major uncertainties in existing models and the interpretation of empirical data [11, 12].

The transport of gases across an air-water interface is commonly described as a diffusive flux that can be parameterized as the product of the gas transfer velocity ($k$) and the difference between dissolved gas concentration at the water surface and the atmospheric equilibrium concentration. The magnitude of $k$ is related to near-surface turbulence at the water side of the interface [13, 14] and is controlled by different hydrodynamic forcing mechanisms, most of them hydrometeorological [15, 16]. Traditionally, $k$ has been estimated as a function of wind speed through empirical models [9, 17, 18], and surface buoyancy flux during convective cooling [19, 20]. Other processes that affect the exchange of gases and near-surface turbulence, such as rainfall, are mostly overlooked, although they may play an important role in gas exchange [4].

At the mechanistic level, the eddy cell model for surface renewal proposed by [13], referred to as the surface renewal model throughout the manuscript, has provided a scaling relationship for gas transfer velocity based on near-surface turbulence, which led to a universal relationship between the gas transfer velocity and the dissipation rate of turbulent kinetic energy near the water surface [14, 21]. The surface renewal model has been validated for a wide range of environmental forcing conditions [16].

Field studies in lakes and reservoirs located at different latitudes have shown that rainfall events can greatly impact the gas transfer velocity [5, 6]. Laboratory studies have found that rainfall significantly increases (by a factor of 3 to 40) air-water gas exchange, particularly at low to moderate wind speeds [10, 18, 22–27]. There is a consensus that gas exchange at the water-atmosphere interface is significantly affected by rainfall. Intuitively, turbulence is believed to be responsible for this effect [10, 26], and growing evidence suggests that rainfall significantly increases near-surface turbulence in marine [10, 24, 27] and freshwater [25] environments. However, the few studies that have investigated this further reported contrasting results. [27] attributed 80% of the reaeration during a rainfall event to turbulence, while [23] concluded that turbulence is the main mechanism affecting $k$ during rainfall events without having direct observations of turbulence. Conversely, other studies have found no significant contribution of rainfall to measured dissipation rates of turbulent kinetic energy [28, 29]. The remaining studies found that rainfall enhances near-surface turbulence, but the turbulence was independent of the rain rate under the investigated experimental conditions [10, 11]. Based on the above results, it has been challenging to establish a consistent relationship between turbulence and rain rate that can be applied in biogeochemical models. Additionally, more emphasis has been given to high rainfall rates (24–190 mm h$^{-1}$ [10, 11, 25, 28, 29], overlooking low to moderate rainfall intensities (rainfall rate < 25 mm h$^{-1}$) which tend to occur more frequently. There is also a bias towards marine environments, in situ or with artificial seawater under laboratory conditions; rain-induced turbulence at the surface of freshwater has rarely been investigated [25].

The influence of rainfall on gas exchange has also been found to become negligible at high wind speeds [25], making assessments of the effect of rainfall in isolation less relevant in regions with high wind speeds. However, in regions where wind speeds are typically low, such as the tropics [4, 25], or in small lakes and reservoirs, the rainfall can be a more important driver of gas exchange, as almost any rain rate can significantly contribute to the gas exchange under low wind conditions ($U_{10} < 5$ m s$^{-1}$) [25]. When using rain rate as a scaling parameter, a

major source of uncertainty comes from the distribution and heterogeneity of raindrop sizes and fall velocities. The assessment of such heterogeneity in both laboratory and environmental conditions requires complex and extensive measurements [10]. Depending on drop size, raindrops reach their terminal fall velocity after a free-fall distance of ~20 m [30], which challenges laboratory setups. As it was shown that rain-induced air-water gas transfer also correlates with the kinetic energy flux of the rain [10, 22–24], the uncertainties related to drop sizes and fall velocities might be mitigated by this quantity [22].

Rain-induced turbulence has been typically characterized via high-resolution spatial mapping of turbulent flow velocities near the water surface [10, 11, 31], or single-point velocity observations at a fixed depth [10, 25, 28, 32]. Spatially-resolved observations of turbulent flow fields are facilitated by particle image velocimetry (PIV) measurements, which have become an affordable, high-resolution technique in laboratory studies due to the advances in camera technology, computer power and the availability of open-source software for processing [33, 34]. However, this technique has mostly remained unexplored in the study of rain-induced turbulence. [11] performed PIV measurements to gain physical understanding of rain-induced turbulence. Their setup, however, focused on high rain rates (40, 100, and 190 mm h$^{-1}$), and thus did not cover the lower end of the wide range of rain rates that occur under environmental conditions.

In this study, we addressed some limitations still present in the quantitative understanding of the effect of rain rate on near-surface turbulence and on the resulting gas transfer velocity in freshwater. We performed controlled laboratory experiments over a wide range of rain rates (7 to 90 mm h$^{-1}$) and estimated gas transfer velocities from high-resolution measurements of $O_2$ concentration, while rain-induced turbulence was characterized based on PIV measurements. Based on these measurements, we elucidate empirical and mechanistic relationships between the gas transfer velocity and near-surface turbulence in dependence on the rain rate and kinetic energy flux. The general applicability of the observed relationship between gas transfer velocity and rain intensity is discussed in the context of surface renewal theory and previous empirical studies.

## Materials and methods

### Experimental setup

Our experimental setup consisted of a custom-built rain generator, which was mounted ~20 m above a rain-collecting aquarium (Fig 1). To exclude the effects of wind and solar radiation, the experiments were conducted in a closed tower (a hose-drying tower of a local fire brigade, S1 Fig) whose height (> 20 m) allowed the raindrops to reach their terminal velocity. Before each experimental series, the dissolved oxygen ($O_2$) concentration in the aquarium water was reduced to about 50% of its atmospheric equilibrium concentration. During experiments with varying rainfall intensity, we observed the reaeration rate of the water using an oxygen mass balance, as well as the turbulent velocity fields near the water surface (Fig 1). In addition, we measured the rain rates and estimated the size and fall velocity of the raindrops.

The rain generator consisted of a frame ($1 \times 1 \times 0.05$ m) with a bottom made of a perforated plexiglass plate (3 mm of thickness). The footprint area of the rain generator exceeded the surface area of the aquarium on all sides, ensuring a homogenous distribution of rain at the water surface. The plate had 580 holes with a diameter of 0.8 mm and with regular spacing. The rain rate was varied by i) controlling the water level above the plate, and ii) changing the number and size of open holes in the plate. The rain generator was manually fed with tap water stored at ambient temperature in an open tank. By varying the water level, we modified the frequency of drop formation at each hole, with a higher level resulting in a higher frequency of droplets

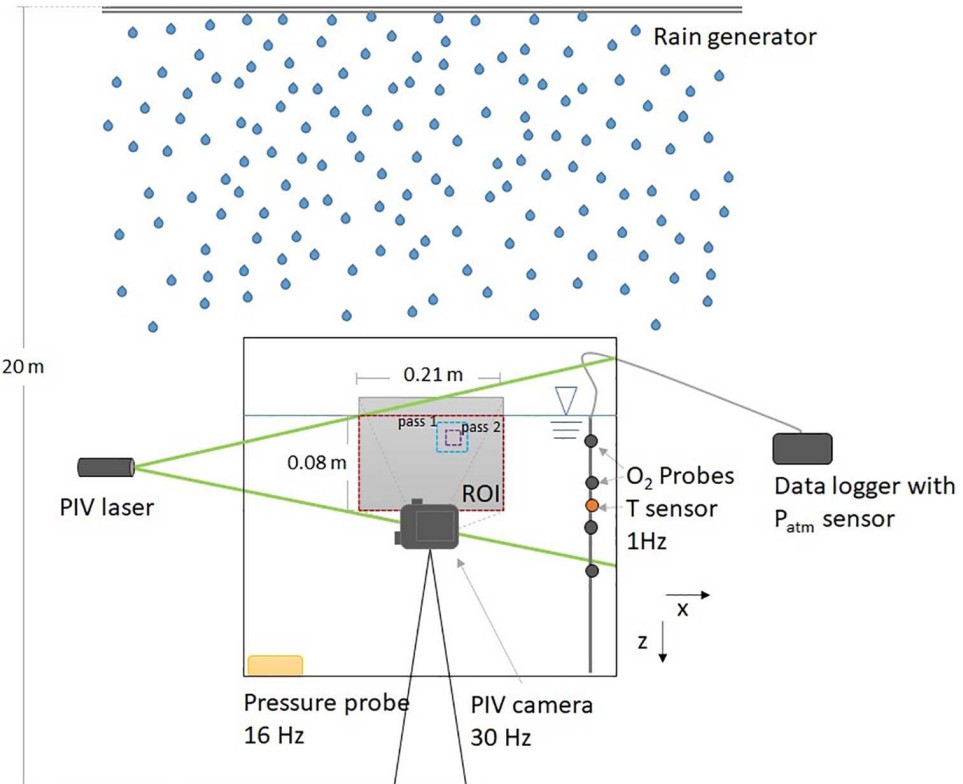

**Fig 1. Schematic of the set-up and instrumentation used in the experiments (not to scale).** The aquarium (0.5 × 0.5 × 0.5 m) was located ~20 m below the rain generator. Particle image velocimetry (PIV) was used to characterize the turbulent flow field in the water. For this, microscopic seeding particles were illuminated from the side using a laser light sheet and observed with a camera through the front window. The field of view of the PIV camera (21 × 12.5 cm on average) is shown as a grey rectangle, the region of interest (ROI) of the PIV measurements (21 × 8 cm on average) is marked by the red dashed line and the size of the interrogation areas is exemplified by blue (pass 1) and purple (pass 2) dashed lines. Four sensors for dissolved oxygen ($O_2$ probes) were used to establish an oxygen mass balance in the aquarium to estimate the gas transfer velocity. A pressure sensor located at the bottom of the aquarium was used to estimate the rain rate from the temporal increase in water level. Atmospheric pressure ($P_{atm}$) was recorded by the oxygen and temperature data logger.

and a higher overall rain rate. Some holes were closed with adhesive tape and subsequently reopened by piercing the tape with a needle (0.4 mm diameter) or by removing the tape to revert to the original hole size. Open and closed holes were always homogeneously distributed across the plate (S2 Fig). During the experiments, we adjusted the number of open holes to finally obtain a range of rain rates between 7 and 90 mm h$^{-1}$, which correspond to the most frequent rain rates observed at a tropical freshwater reservoir, with many occurring at low wind speeds of < 3 m s$^{-1}$ (S3 Fig). Similar ranges were reported for rain rates observed in the tropical and temperate zones [3, 35, 36].

## Measurements and experiments

The aquarium that received the rainwater had a volume of 125 L (50 × 50 × 50 cm) and was initially filled to a height of ~42 cm. The instrumental setup inside and outside of the aquarium facilitated estimating the gas transfer velocity, turbulent energy dissipation rates in water, and the rain rate (Fig 1).

Each experimental series of measurements consisted of a 15 min period without rain, followed by several 15 min runs with constant rain rates (rain periods). During the runs, we maintained the same configuration (size and number of open holes) but varied the water level in the rainfall generator to vary the rain rate among runs. For the following experimental series, the configuration of the plate was changed, and the procedure was repeated. Rain rates ($R$ in mm h$^{-1}$) were estimated for each run as the mean rate of change of the water level in the aquarium, which was monitored using a high-resolution pressure probe (Duet, RBR Inc., sampling at 16 Hz with 0.2 mm of depth resolution). Examples of time series of RBR pressures are shown in S4 Fig.

For estimating gas transfer velocities, we monitored the dissolved oxygen concentration in water with four fiber-optic sensors (FireStingO2, PyroScience GmbH), distributed vertically at 3, 13, 23 and 33 cm depth from the water surface. The sensors recorded time series at 1 Hz frequency during the 15 min duration of each experimental run. The water temperature and atmospheric pressure ($P_{atm}$) were measured at the same frequency. As the biological activity may affect the oxygen balance, we used tap water and carefully checked the results of the no-rain periods to discard possible biological activity increasing or decreasing dissolved oxygen. As the oxygen in water was mostly constant during no-rain periods, and it only increased when the rain started (S5 Fig), it was assumed that biological activity, if present, was low and negligible in the context of our mass balance and the short duration of the experimental runs. The water temperature varied between the experimental series (from 8.4 to 18.5˚C) but remained relatively constant during the individual runs (Table 1).

To visualize the turbulent flow field, seeding particles (polyamid, 20 μm diameter) were added to the water in the aquarium and illuminated with a continuous-wave laser light sheet. We used a green (532 nm), continuous-wave line laser (450 mW; Inline HP, MediaLas Electronics GmbH) to illuminate a vertical plane in the center of the aquarium. The

**Table 1. Summary of experimental results.**

| Run | $R$ [mm h$^{-1}$] | $T$ [˚C] | $Sc_{O2}$ | $\epsilon$ [W kg$^{-1}$] | $k_{600}$ [cm h$^{-1}$] | $k_{600\_mod}$ [cm h$^{-1}$] | $F_{KE}$ [W m$^{-2}$] |
|---|---|---|---|---|---|---|---|
| 1 | 6.90 | 16.9 ± 0.054 | 623 | $3.10 \times 10^{-8}$ | 2.44 | 11.6 | 0.0779 |
| 2 | 8.05 | 17.1 ± 0.055 | 615 | $3.76 \times 10^{-8}$ | 6.43 | 12.4 | 0.0905 |
| 3 | 10.3 | 9.48 ± 0.030 | 927 | $3.09 \times 10^{-8}$ | 4.69 | 14.6 | 0.116 |
| 4 | 13.5 | 9.35 ± 0.027 | 933 | $9.74 \times 10^{-8}$ | 16.2 | 16.4 | 0.152 |
| 5 | 16.0 | 9.13 ± 0.737 | 945 | $1.62 \times 10^{-7}$* | 52.7 | 17.8 | 0.180 |
| 6 | 16.2 | 8.59 ± 0.020 | 973 | $1.21 \times 10^{-7}$ | 25.2 | 20.2 | 0.237 |
| 7 | 21.1 | 8.36 ± 0.016 | 979 | $9.97 \times 10^{-8}$ | 17.5 | 17.9 | 0.182 |
| 8 | 19.8 | 8.49 ± 0.021 | 986 | $9.74 \times 10^{-8}$ | 17.9 | 19.7 | 0.222 |
| 9 | 25.0 | 16.8 ± 0.062 | 624 | $9.68 \times 10^{-8}$ | 15.5 | 20.6 | 0.281 |
| 10 | 26.0 | 11.0 ± 0.010 | 852 | $1.36 \times 10^{-7}$ | 20.8 | 21.8 | 0.293 |
| 11 | 28.8 | 17.4 ± 0.087 | 607 | $9.62 \times 10^{-8}$ | 28.8 | 21.8 | 0.323 |
| 12 | 39.4 | 11.1 ± 0.013 | 849 | $3.14 \times 10^{-7}$ | 30.1 | 26.2 | 0.443 |
| 13 | 48.9 | 17.8 ± 0.146 | 593 | $2.62 \times 10^{-7}$ | 25.3 | 27.6 | 0.550 |
| 14 | 88.9 | 18.5 ± 0.220 | 572 | $3.42 \times 10^{-7}$ | 41.8 | 35.9 | 0.999 |

$R$ is the rain rate, $T$ is the water temperature and $Sc_{O2}$ is the Schmidt number of oxygen at temperature $T$. $\epsilon$ is the rain-induced turbulent dissipation rate from the PIV estimated at 7.5 cm depth. $k_{600}$ and $k_{600\text{-}mod}$ are the estimated (Eq (5)) and modeled (Eq (13)) gas transfer velocities, respectively. $F_{KE}$ is the estimated kinetic energy flux of rain.

* Marks an outlier that caused the run with 16 mm h$^{-1}$ to be excluded from subsequent analysis

dynamic distribution of the light scattering seeding particles was observed by recording videos with a consumer-grade camera (GoPro Hero4, GoPro Inc.) located at ~13 cm from the front face of the aquarium, with a frame rate of 30 fps (30 Hz of temporal resolution) and a resolution of 1920 × 1080 pixels. The field of view covered an area of 21 × 8 cm below the water surface and a portion of the air side of 3.5 cm height (Fig 1 and S6 Fig). Videos for Particle Image Velocimetry (PIV) measurements were performed for 3 min during each run. Additional videos were made without rain at the beginning of each experimental series, to characterize background turbulence levels. For metric calibration of the videos, a short (~3 s) video was recorded before each series while placing a calibration target (0.5 × 0.5 cm checkerboard pattern) in the laser light sheet. Simultaneously to the PIV videos, single-point measurements of 3-dimensional flow velocities were carried out with an acoustic Doppler velocimeter (ADV; Vector, Nortek A/S) sampling at 32 Hz. The ADV instrument was placed in the aquarium upside-down with the center of the acoustic sampling volume (~ 15 mm diameter and 14 mm length) located at 3 cm depth. The ADV velocity range was set to 0.3 m s-1, ensuring the resolution of vertical and horizontal velocities up to 0.23 and 0.81 m s$^{-1}$, respectively.

The rain rate and gas transfer velocities were estimated for the 15-minute duration of each experimental run. However, during the 3-minute videos for the turbulence measurements by PIV, the water level increased by a maximum of 4.5 mm (for 90 mm h$^{-1}$), which represents only 1% of the water column depth, therefore, the vertical axis was fixed at the initial water level and the water level change was neglected for the dissipation rate estimates.

## Fall velocity, drop sizes and kinetic energy flux of raindrops

The final fall velocity and sizes of raindrops were estimated by placing a planar light at the location in the aquarium and taking videos of the falling raindrops between the light and the camera (backlight illumination). The videos were recorded with a Sony RX100 IV camera at a frame rate of 1000 fps and a resolution of 1920 × 1080 pixels. A short video of a calibration target was recorded using the same settings. For estimating fall velocity, the recorded videos were first overlaid to the image of the calibration target, and the number of consecutive frames in which individual drops could be observed was counted alongside the vertical distance they traveled. Estimates of drop sizes were only performed for a plate configuration with open holes of 0.8 mm. For drops generated using syringe needles (diameters of 0.4 mm), we expected that the resulting drop sizes would be comparable to those under the configuration with the pierced tape in the plate [32]. To validate the observed fall velocities, the measured drop sizes were used to estimate the drop's terminal velocity using the established model by van [37].

The kinetic energy flux of the rain ($F_{KE}$, in J m$^{-2}$ s$^{-1}$) was calculated according to the expression proposed by [22] as follows:

$$F_{KE} = \frac{1}{2}\rho R V^2 \tag{1}$$

where $\rho$ is the water density, $V$ is the average fall velocity of the drops and $R$ is the rain rate (converted to m s$^{-1}$).

## Gas transfer velocity

Estimates of the gas transfer velocity of oxygen ($k_{O_2}$) were obtained from dissolved oxygen (O$_2$) profiles, following a mass balance approach. The total rate of change of dissolved oxygen concentration in the aquarium was equated to the sum of the diffusive flux at the water surface

($F_{dif}$) and the oxygen flux from rain ($F_{rain}$). Solving for the diffusive flux as follows:

$$F_{total} = \sum_{i=1}^{4} \frac{dC_i}{dt} \Delta h_i \qquad (2)$$

where $dCi/dt$ is the mean rate of change of dissolved $O_2$ concentration measured by each $O_2$ probe (in $\mu$mol L$^{-1}$ d$^{-1}$) and $\Delta h_i$ is the representative layer thickness for each sampling depth (in m). The oxygen probes were fixed relative to the bottom of the aquarium and the inflow of rainwater raised the thickness of the upper layer. Therefore, $\Delta h_i$ was constant for all the probes, except the upper one, for which $\Delta h_i$ changed with time according to the rainfall rate.

We assumed that the rainwater is saturated with oxygen at the same temperature as the aquarium water after being at rest for several hours and open to the atmosphere, then the oxygen flux associated with rain ($F_{rain}$, in mmol m$^2$ d$^{-1}$) in the aquarium was as follows:

$$F_{rain} = RC_{eq} \qquad (3)$$

where $C_{eq}$ is the $O_2$ concentration in equilibrium with the atmosphere at the given water temperature (in mmol L$^{-1}$) [38], and $R$ was converted to volumetric units (mm h$^{-1}$ = L h$^{-1}$).

The gas transfer velocity of oxygen at in-situ temperature ($k_{O_2}$ in m d$^{-1}$) was estimated using Fick's first Law as follows:

$$k_{O_2} = \frac{F_{dif}}{(C_w - C_{eq})} \qquad (4)$$

where $C_w$ is the temporarily averaged $O_2$ concentration measured by the uppermost sensor (in mmol L$^{-1}$). $k_{O2}$ was normalized to $k_{600}$, the gas transfer velocity of $CO_2$ at 20°C as follows:

$$k_{600} = k_{O_2} \left( \frac{600}{Sc_{O_2,T}} \right)^{-n} \qquad (5)$$

where $Sc_{O2}$ is the temperature-dependent Schmidt number of oxygen (calculated following [39] ($Sc_{O2} = v/D_{O2}$, with $v$ being the viscosity of water and $D_{O2}$ the diffusion coefficient of oxygen in water). We used the recommended exponent for wavy surfaces ($n = 0.5$) [40].

## Particle image velocimetry

We extracted frame sequences (in bitmap format) from each video that were subsequently converted to greyscale images. The pre-processing of the image sequences and PIV analyses were performed using the open-source program PIVlab (v. 2.50) [34, 41]. The imported images were first preprocessed to increase and homogenize the contrast and sharpness of the laser-illuminated seeding particles, and to remove the image background. The region of interest (ROI) for PIV analysis was selected just below the craters formed by the raindrops to the maximum resolved depth (Fig 1 and S6 Fig). One frame of the calibration video was used for conversion from pixels to metric units, with pixel size varying between 0.11 and 0.13 mm among experimental series. The images were corrected for lens distortion and calibrated in Matlab 2019a using the Single Camera Calibrator App (reprojection error between 0.47 and 0.98 px). Discrete Fourier Transform correlation with multiple passes and deforming windows was used for PIV processing and two passes with 50% step size were applied, starting with an interrogation area of 128 × 128 pixels and decreasing to 64 × 64 pixels. Subpixel estimation was done using a Gaussian 2.3-point fit. The final spatial resolution of velocity vectors was 3.9 ± 0.2 mm.

Based on the results obtained by [42] and to be conservative, we consider 0.1 pixel accuracy ($1.3 \times 10^{-5}$ m) in accordance with [11], which translated into velocities of $4 \times 10^{-7}$ m s$^{-1}$ for a temporal resolution of 30 s$^{-1}$ and 1 pixel being 0.00012 m (see details of the estimated accuracy of the PIV measurements in SI).

We used a standard deviation filter with a threshold of a factor of eight, as well as a local median filter with a threshold of 3 for removing velocity outliers and interpolated the missing data after filtering. PIVlab uses a boundary value solver for interpolation, which was originally developed for reconstructing images with missing information. The approach provides an interpolation that is generally smooth, and over larger regions with missing data, it will tend towards the average of the boundary velocities, which prevents overshooting [43].

The PIV analysis provided two-dimensional velocity distributions with 3.9 mm spacing for $n \sim 5400$-time steps (corresponding to the total number of frames in each video of $\sim 3$ min duration and recorded at 30 fps). The two resolved velocity components (horizontal velocity $u$ and vertical $w$) were despiked using a modified phase-space method [44]. Mean values of the velocity components ($\bar{u}$ and $\bar{w}$) were calculated as the time average for each cell of the velocity fields (in m s$^{-1}$). Turbulent velocity fluctuations were calculated based on Reynolds decomposition by subtracting the mean velocities from the instantaneous flow velocities as follows:

$$u' = u - \bar{u} \tag{6}$$

$$w' = w - \bar{w} \tag{7}$$

## Dissipation rates of turbulent kinetic energy

Viscous dissipation rates of turbulent kinetic energy ($\epsilon$) result from velocity gradients along all three spatial dimensions. We used a "direct" estimate of the dissipation rates ("PIV_dir") from the four components of the velocity gradients that are resolved in two-dimensional PIV measurements. According to the expression in [45], turbulent dissipation rates were calculated (in W kg$^{-1}$ or m$^2$ s$^{-3}$) as follows:

$$\epsilon = 3v\left[\left(\frac{\partial u'}{\partial x}\right)^2 + \left(\frac{\partial w'}{\partial z}\right)^2 + \left(\frac{\partial u'}{\partial z}\right)^2 + \left(\frac{\partial w'}{\partial x}\right)^2 + 2\left(\frac{\partial u'}{\partial z}\frac{\partial w'}{\partial x}\right) + \frac{2}{3}\left(\frac{\partial u'}{\partial x}\frac{\partial w'}{\partial z}\right)\right] \tag{8}$$

where $x$ and $y$ denote horizontal ($u$) and vertical ($w$) coordinates, respectively. Kinematic viscosity ($v$) was in m$^2$ s$^{-1}$. Time-average values of the two-dimensional distributions of dissipation rates ($\epsilon$) were calculated as averages of log-transformed values [46] for all rain experiments. Finally, dissipation rates were log-averaged horizontally to obtain a mean vertical profile of the turbulent dissipation rate ($\epsilon$) for each run.

Turbulent energy dissipation rates were additionally estimated by the inertial subrange method ($\epsilon_{spec}$) using velocity time series measured by PIV ("PIV_spec") and by the ADV. The approach is based on the assumptions of isotropy and fully developed turbulence, for which the wavenumber spectrum of the turbulent velocity fluctuations follows a universal form [47]:

$$S = \alpha \frac{18}{55} \epsilon_{spec}^{2/3} \kappa^{-5/3} \tag{9}$$

where $S$ (in (m s$^{-1}$)$^2$ (rad m$^{-1}$)$^{-1}$) is the one-dimensional wavenumber spectrum of the turbulent velocity fluctuations, $\alpha$ is taken to be $1.5 \times 4/3$ (factor for the vertical direction), and $\kappa$ is the wavenumber (frequency divided by mean velocity). The wavenumber spectra (PIV_spec) were estimated for each run at 3 cm depth to be consistent with ADV results.

## Scaling of rain-induced gas transfer

As near-surface dissipation rates from wind shear stress have been shown to follow a universal power-law decline with increasing depth [48], we tested a power-law approach for describing the vertical attenuation of rain-generated turbulence:

$$\epsilon_z = a\,z^{-b} \tag{10}$$

where $\epsilon_z$ can be estimated for any water depth $z$ below the viscous sublayer at the water surface. For all runs, dissipation rate profiles were fitted to Eq (10) to obtain a common exponent $b$, and a function that describes the dependence of coefficient $a$ on the rain rate ($R$). Thus, an empirical function $\epsilon_{Rain} = f(z, R)$ was obtained with $\epsilon_{Rain}$ in W kg$^{-1}$, $z$ in m and $R$ in mm h$^{-1}$.

In order to explore a mechanistic approach for modeling the gas transfer velocity ($k$, in m s$^{-1}$) in response to rain-induced turbulence, the above scaling of the turbulent dissipation rate as a function of the rain rate was combined with the surface renewal model (Eq (11); Lamont & Scott, 1970) as follows:

$$k = A\,Sc^{-n}(\epsilon\nu)^{0.25} \tag{11}$$

To estimate the dimensionless empirical coefficient $A$, linear regressions were made between the estimated values of $k_{600}$ and the expression $600^{-n}(\epsilon\nu)^{0.25}$. The regressions were made for dissipation rates measured at the same depth, which we obtained by interpolation of the mean vertical profiles ($\epsilon_z$) between 0.5 to 7.5 cm depth and with a vertical resolution of 0.5 cm. Interpolated profiles were computed for the rain periods and the dissipation rates resulting from rain ($\epsilon_{Rain}$) were estimated. Then, we analyzed the dependence of the gas transfer velocity on rain rate ($R$) and obtained 15 linear regressions corresponding to different depths at which dissipation rates were measured ($z_1 = 0.5$ cm, $z_2 = 1.0$ cm,. . . $z_{15} = 7.5$ cm). The quality of the correlations was assessed using the coefficient of determination ($r^2$), and the best correlation was selected for the final model ($k_{600\_mod}$) by replacing $\epsilon$ in Eq (11) with the function $\epsilon_{Rain} = f(z, R)$.

## Results

### Overview

The different configurations of the rain generator resulted in rain rates between 6.9 and 88.9 mm h$^{-1}$, (Table 1). Both $k_{600}$ and dissipation rates obtained at a rain rate of 16.0 mm h$^{-1}$ presented outliers in all subsequent analyses (Table 1). This run was excluded from the following analyses of dissipation rates and from the $k_{600}$ analysis.

In the absence of rain, the oxygen concentration in the aquarium did not change significantly while significant increases in oxygen concentration were observed with time in the presence of rain (S5 Fig). Gas transfer velocities were estimated using Eq (5) for rain periods (Table 1). During all experiments with rain, the dissolved oxygen concentrations increased nearly linearly with time ($r^2 > 0.90$ for most of the measurements, $p < 0.0001$), with similar rates at all sampling depths and with increasing slopes for increasing rain rates (S5 Fig). The in-situ gas transfer velocity $k_{O2}$ was strongly correlated with the rain rate ($r^2 = 0.87$) (Fig 2a). In contrast, the normalized gas transfer velocity ($k_{600}$), which ranged from 3.0 to 46.5 cm h$^{-1}$ (Table 1), showed a slightly weaker correlation with the rain rate when all data were included ($r^2 = 0.76$) (Fig 2b).

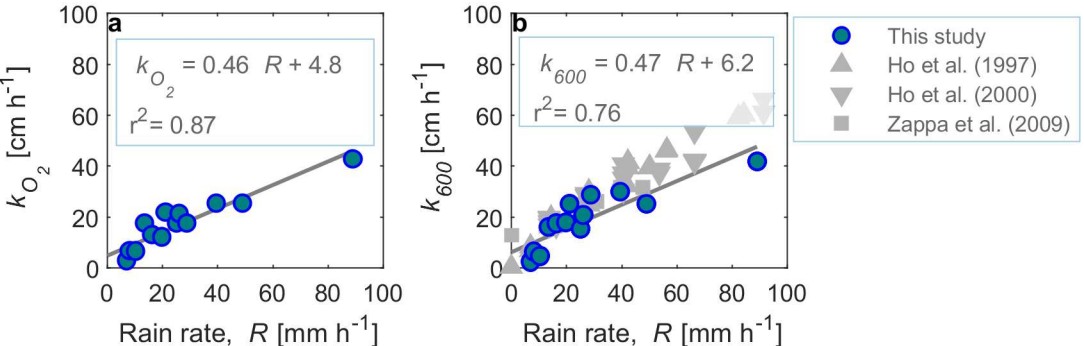

**Fig 2. Gas transfer velocity versus rain rate ($R$) with linear regressions (solid lines) and the resulting equations shown as legends. a)** Gas transfer velocity for oxygen at in situ temperature ($k_{O_2}$), **b)** Normalized gas transfer velocity ($k_{600}$) and comparison to previous studies [10, 22, 23]. The data from [10] were fitted with a linear regression to obtain the expression $k_{600} = 0.42R + 13.3$ ($r^2 = 0.97$).

## Energy dissipation rates

Time-averaged turbulent dissipation rates $\epsilon_{t\_avg}$ were homogeneously distributed along the horizontal direction but showed strong vertical gradients (S7 Fig). They were largest near the water surface and decreased by about two orders of magnitude toward the lower part of the field of view at ~7 cm depth. Higher rain rates were associated with higher energy dissipation rates. Thus, $\epsilon_{t\_avg}$ varied vertically between ~$10^{-5}$ and ~$10^{-7}$ W kg$^{-1}$ for rain rates $R \leq 25$ mm h$^{-1}$, and from ~$10^{-4}$ to ~$10^{-6}$ W kg$^{-1}$ for $R > 25$ mm h$^{-1}$ (S8 Fig).

The depth dependence of mean turbulent dissipation rates ($\epsilon_{Rain}$) could be well described by power-law functions for all investigated rain rates ($r^2 \geq 0.98$, S8 Fig). The exponent $b$ (Eq (10)) was relatively constant for all rain rates ($b = 1.99 \pm 0.33$ mean ± std.). The coefficient $a$ in Eq (10) was related to the rain rate ($R$) by a power-law function with an exponent of 1.8 ($r^2 = 0.80$, Fig 3a). Combining both relationships resulted in the following empirical scaling of the dissipation rate ($\epsilon_{model}$ in W kg$^{-1}$) as a function of the rain rate ($R$ in mm h$^{-1}$) and depth ($z$ in m):

$$\epsilon_{model}(R, z) = 2.28 \times 10^{-12} R^{1.8} z^{-2.0} \tag{12}$$

In general, modeled and measured dissipation rates were in good agreement (Fig 3b). Particularly the decline of dissipation rates by two orders of magnitude over the topmost 7 cm of

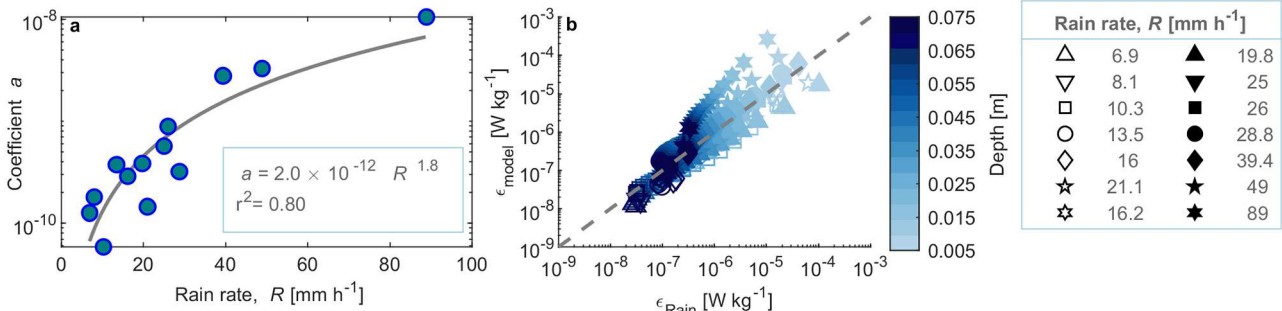

**Fig 3. Details of the dissipation rate modeling process. a)** Power-law relationship between the coefficient $a$ (Eq (9)) and the rain rate ($R$) (solid line). **b)** Modeled versus measured dissipation rates for all evaluated rain rates ($R$) (symbols) and at all evaluated sampling depths (colorbar). The dashed line shows a 1:1 relationship.

the water column was consistent and well described by the model. The overall model showed the best agreement for moderate rain rates around 30–40 mm h$^{-1}$, while the magnitude of dissipation rates tended to be underestimated for the lowest rain rates (Fig 3b).

The inertial subrange method using the data from the ADV was used to validate the PIV results from the direct method ("PIV_dir") from Eq (8). ADV dissipation rates ($\epsilon_{ADV}$) ranged from $1.2 \times 10^{-8}$ to $5.2 \times 10^{-6}$), showing a general consistency with the magnitude of the PIV-based estimates in the rain rate range between 7 and 21 mm h$^{-1}$. For higher rain rates, the ADV results were about one order of magnitude larger than the ones from the PIV. We additionally estimated the dissipation rates by the inertial subrange method using the PIV results ("PIV_spec") and the results from both methods (PIV_dir and PIV_spec) were comparable to each other (S1 Table).

## Scaling of the gas transfer velocity

We compared the normalized gas transfer velocities ($k_{600}$) observed at different rain rates ($n$ = 14) against those predicted from turbulent dissipation rates by the surface renewal model (Eq (11)). The empirical coefficient $A$ in the surface renewal model was estimated separately for all sampling depths of dissipation rates by linear regression (S9 Fig). The correlations were systematically weaker near the water surface and became stronger toward $\geq$6 cm depth ($r^2$ ~0.50). The best correlation with measured gas transfer velocities was obtained from dissipation rates measured at 7.5 cm depth ($r^2$ = 0.52), with an estimated value of $A$ = 2.31 (Fig 4),

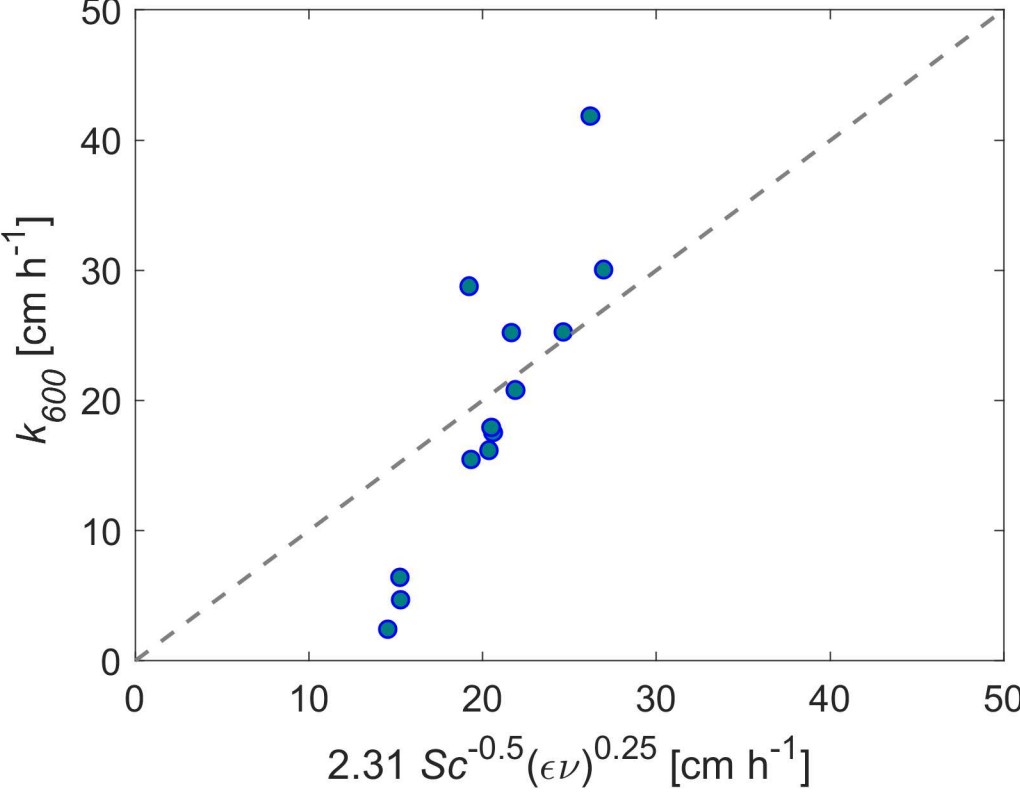

**Fig 4. Normalized gas transfer velocities $k_{600}$ versus surface renewal model (Eq (11)) for energy dissipation rates sampled at $z$ = 7.5 cm depth.** At this depth, we estimated the value of the coefficient $A$ as 2.31, $Sc$ is the Schmidt number (here equal to 600). The dashed line shows a 1:1 relationship.

such that Eq (13) becomes as follows:

$$k_{600\_mod}[\mathrm{m\ s^{-1}}] = 2.31 \cdot 600^{-0.5}(\epsilon_{model}\nu)^{0.25} \tag{13}$$

To relate the gas transfer velocity to rain rate ($R$), we replaced the measured dissipation rate in the surface renewal model by the function $\epsilon = f(z, R)$ given in Eq (12), to obtain a model of $k_{600}$ (in cm h$^{-1}$) as a function of the rain rate ($R$ in mm h$^{-1}$) and kinematic viscosity ($\nu$ in m$^2$ s$^{-1}$) at 7.5 cm depth as follows:

$$k_{600\_mod}[\mathrm{cm\ h^{-1}}] = 152\,R^{0.45}\nu^{0.25} \tag{14}$$

Gas transfer velocities modeled from rain rates varied from 11.6 to 35.9 cm h$^{-1}$ (Table 1). Good agreement was found when the model ($k_{600\_mod}$) was linearly correlated with all accepted data of the measured $k_{600}$ (Fig 5) ($r^2 = 0.85$, $n = 13$, $p < 0.0001$).

## Kinetic energy flux

The drop size was estimated to be 5.7 ± 1.1 mm ($n = 103$). Since the drop sizes were only estimated from the configuration with the largest open holes (0.8 mm), it is assumed that according to the needle and droplet diameters measured by [32], droplets of ~2.1 mm diameter were produced all the time by at least 50% of the holes according to the configurations we used in the rain generator. The fall velocity of the raindrops above the water surface was estimated from observations of 6 individual drops to be 8.3 ± 2.5 m s$^{-1}$ (mean ± std.). This velocity was

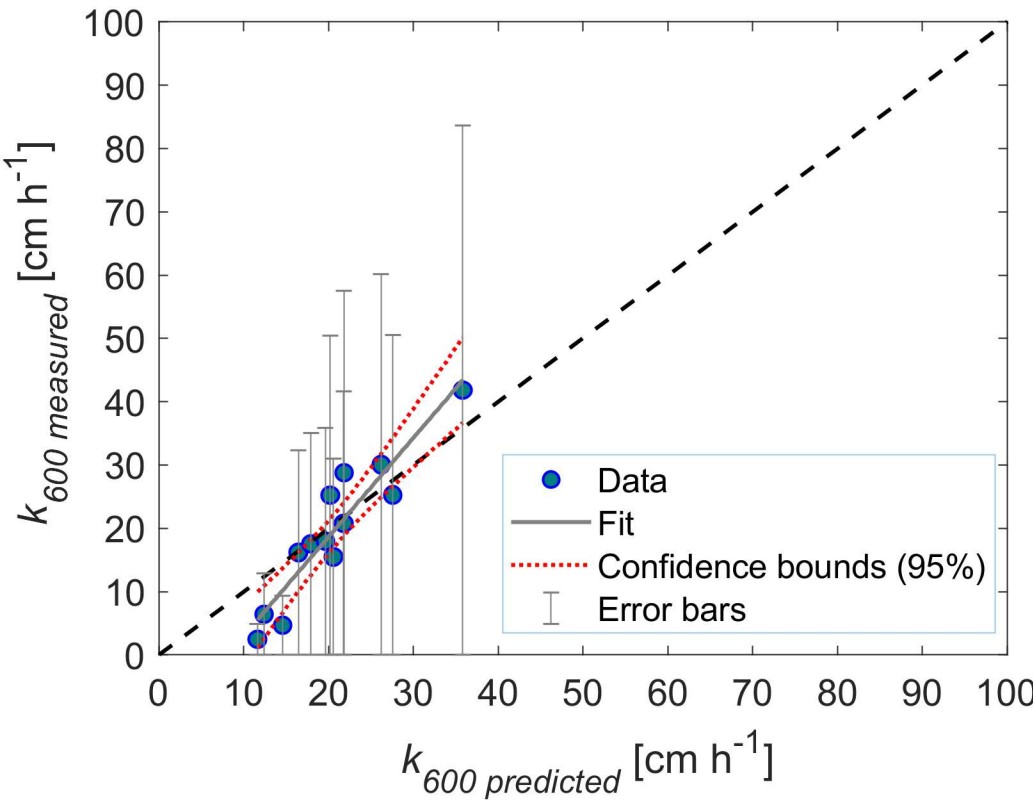

**Fig 5. Measured ($k_{600}$) versus predicted ($k_{600\_mod}$) gas transfer velocities (symbols) and a 1:1 plot (dashed line).** The slope of a linear regression (grey line) was 1.6 ± 0.18 ($r^2 = 0.85$, $n = 13$, $p < 0.0001$). Error bars represent the uncertainty of a factor of 2 in the measured dissipation rates.

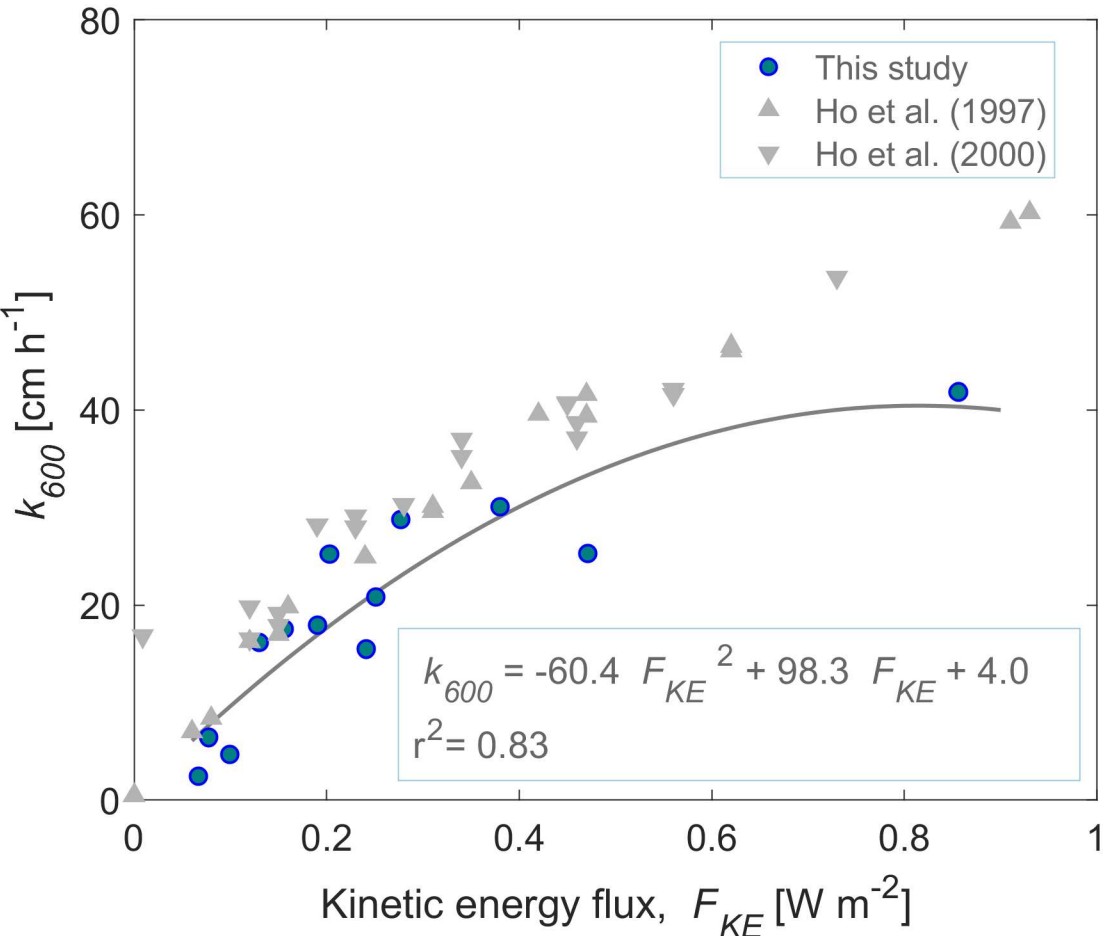

**Fig 6. Observed gas transfer velocity ($k_{600}$) versus kinetic energy flux of rain ($F_{KE}$).** The solid line shows a second-order polynomial fit according to the equation provided in the legend. Results from [22, 23]. The data from were taken from [22] and fitted to a polynomial curve to get the expression $k_{600}$ = -30.6 $F_{KE}^2$ + 91.7 $F_{KE}$ + 3.47 ($r^2$ = 0.99) (S10 Fig). The data from [23] were taken from [10] and the polynomial curve was fitted to get the expression $k_{600}$ = -26.9 $F_{KE}^2$ + 84.7 $F_{KE}$ +5.91 (S11 Fig).

found to be consistent with the model of [37]. The resulting $F_{KE}$ ranged from 0.078 to 0.999 W m$^{-2}$ from the lowest to the highest rain rate (Table 1). For comparison with previous studies, we analyzed the relationship between the normalized gas transfer velocity and the kinetic energy flux by fitting a second-order polynomial function to $k_{600}$ as a function of $F_{KE}$ (in W m$^{-2}$) (Fig 6), resulting in:

$$k_{600} \, [\text{cm h}^{-1}] = -60.4 F_{KE}^2 + 98.3 F_{KE} + 4.0 \qquad (15)$$

## Discussion

### Rain-induced near-surface turbulence

Controlled laboratory experiments were performed to characterize rain-induced turbulence in freshwater over a wide range of rain rates. We found a systematic increase in dissipation rates ($\epsilon_{Rain}$) with rain rate (Eq (12)) that was not previously reported [10, 25, 28]. Our study

expanded the range of rain rates considered in previous studies, which were mostly limited to testing fewer and mostly high rain rates: 108 and 141 mm h$^{-1}$ [28]; 0, 30 and 60 mm h$^{-1}$ [25]; 24, 30, 40 and 48 mm h$^{-1}$ [10] to include low to moderate rain ($< 25$ mm h$^{-1}$) and a larger amount of observations (14 runs). Those conditions allowed us to obtain a consistent power-law relationship between dissipation rates and rain rate (Eq (12)).

Turbulent dissipation rates ($\epsilon$) ranged from $3.1 \times 10^{-8}$ to $3.4 \times 10^{-7}$ W kg$^{-1}$ at 7.5 cm depth for rain rates between 7 and 89 mm h$^{-1}$. In general, these values are three orders of magnitude lower than those reported by [10] and one order of magnitude lower than those from [11], both measured within a few centimeters ($\sim$ 5 to 8 cm) from the surface and at similar rain rates. In contrast, at 1 cm depth [11] observed dissipation rates for a rain rate of 39.4 mm h$^{-1}$ that were in close agreement with our measurements ($\sim 3 \times 10^{-5}$ W kg$^{-1}$ versus $\sim 1 \times 10^{-5}$ W kg$^{-1}$, Fig 7). At greater depth, our estimates were up to one order of magnitude lower. This comparison with previous measurements may suggest that the dissipation rates were underestimated in our study. One possible reason could be an insufficient resolution of the current shear at the smallest scales of motion by our PIV setup. [49] showed that almost the entire dissipation takes place at wavenumbers $\kappa$ for which $\kappa\eta < 5$, where $\eta$ is the Kolmogorov microscale of turbulence ($\eta = (v^3/\epsilon)^{1/4}$. Given rain-driven dissipation rates between $\sim 10^{-8}$ and $\sim 10^{-6}$ W kg$^{-1}$, covering the ranges of the PIV and ADV results, the $\eta$ varied between 1 and 3 mm. The smallest spatial scale resolved in our measurements was 3.9 mm yielding a wavenumber ($\kappa$) of 1611 rad m$^{-1}$, so the values of $\kappa\eta$ range from 1.6 to 4.8. Following [50], we estimated that we resolved about 75% of the total dissipation rate under these conditions. Although this represents a systematic bias of our dissipation estimates, it is small compared to the order of magnitude differences in comparison to previous studies.

Additionally, we applied the inertial subrange method which has also been used in previous rain studies to determine $\epsilon$ [10], we found the dissipation rates from both ADV and PIV-based velocity measurements to be largely comparable (S1 Table, Fig 7). ADV-based dissipation estimates were larger for the highest rain rates. These discrepancies are expected because methodological and sampling limitations cause dissipation rates to be generally associated with relatively large uncertainties, and comparisons among different methods often result in agreement within a factor of two [51]. It should be noted that the application of the inertial subrange method to the single-point ADV velocity measurements in our experiment is rather questionable, because of the general lack of a uniform mean flow, which is required for converting frequency to wavenumber spectra [47].

As a most likely reason for the comparably low dissipation rates generated by rain in our study, we note that earlier studies were carried out in the presence of other turbulence sources such as currents and artificially generated waves, which might become dominant over rain-generated turbulence, whereas we studied the effect of rain in isolation. We therefore consider the resulting relationship between dissipation rate and rain rate as being applicable under rainy and low wind conditions.

## Scaling of rain-induced gas transfer

Our results clearly show that rainfall enhances the gas transfer velocity, which is in agreement with previous observations [10, 22, 24–26, 32]. The resulting normalized gas transfer velocities ($k_{600}$) during rain were mostly within the range reported in previous freshwater studies, which tested rain in isolation from other significant sources of turbulence [22, 23, 25]. Our data were in the upper range of the transfer velocities obtained by [25], which could be related to the different fall velocities of the raindrops. The drops generated at 20 m above the aquarium in our study are expected to have reached their terminal velocity [52], whereas in [25] the height of

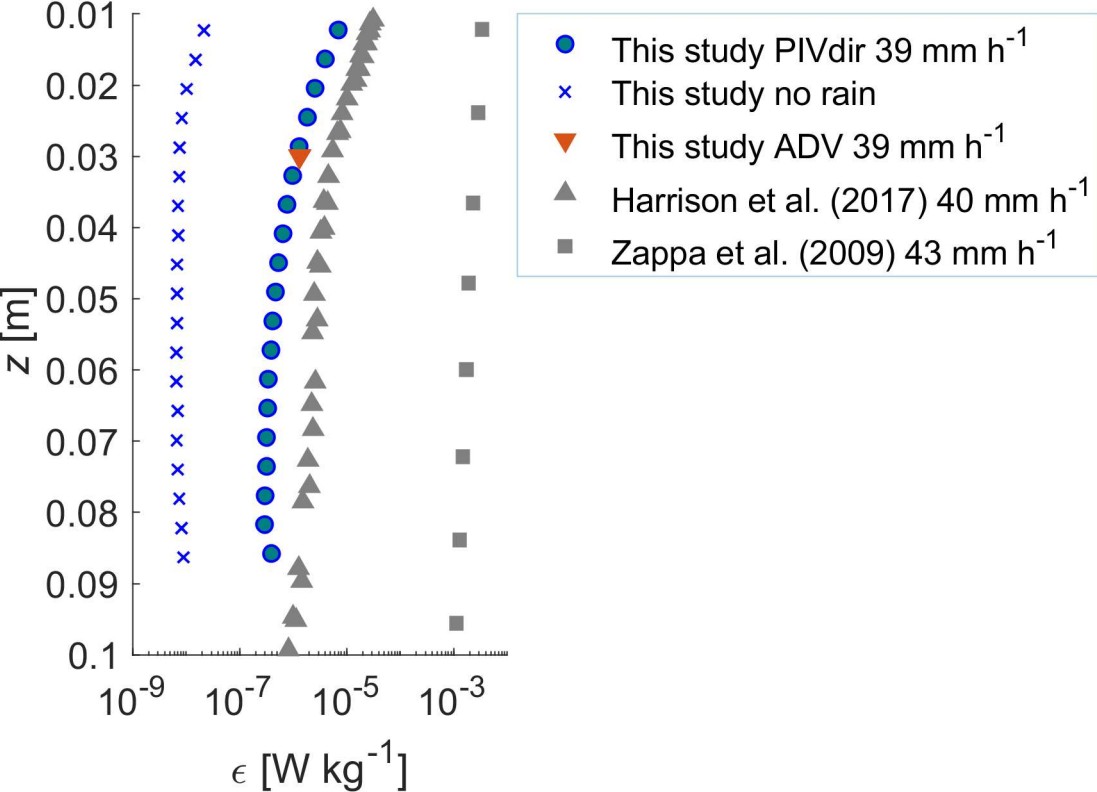

**Fig 7. Comparison of dissipation rate profiles estimated by different methods in the present study, and estimated in other studies for similar rain rates:** PIV-based dissipation rates (Eq 8) for $R$ = 39 mm h$^{-1}$ (blue circles) and in the absence of rain (blue crosses); the single-point dissipation obtained from the ADV data (red triangle); profiles of the first 10 cm obtained by [11] at 40 mm h$^{-1}$ (grey triangles) and data from [10] measured at 43 mm h$^{-1}$ (grey squares).

the rain simulator above the flume was only 2.6 m, leading to smaller fall velocities around 3.13 m s$^{-1}$.

Linear regressions of $k_{600}$ and rain rate $R$ showed strong correlations ($r^2$ = 0.76). However, the oxygen gas transfer velocity ($k_{O2}$) correlated better with $R$ than $k_{600}$ ($r^2$ = 0.87), suggesting additional uncertainty being related to the temperature correction (Schmidt number scaling) of the transfer velocities.

The kinetic energy flux ($F_{KE}$) proposed by [22] as a suitable parameter to explain gas exchange, and that has been used in many subsequent studies [10, 23–25], was well correlated with the rain-induced gas transfer velocity. The resulting magnitude of the $F_{KE}$ was similar to the values reported by [22] and by [10] (Fig 6). Similar to the $k_{600}$—$R$ correlation results, the polynomial expression found in this study for $k_{600}$ as a function of $F_{KE}$ was strong ($r^2$ = 0.83), and the data were in agreement with the empirical fits reported in previous studies [10, 22, 23] for $k_{600}$ < 30 cm h$^{-1}$ (Fig 6).

Mechanistic models of greenhouse gas dynamics and fluxes in aquatic environments are typically based on mass balances, where gas transfer velocities are either obtained from a few solitary measurements or described as a function of mean wind speed [53–55]. In this study, we present a mechanistic method for describing the rain-induced gas transfer.

In line with [10], we found good agreement between observed gas transfer velocities and predictions by the surface renewal model (Fig 5), contributing to the idea of this model as a

possibly unified relationship for the air-water interfacial fluxes in response to a range of environmental forcing conditions. However, the large difference between our dissipation rates and values observed by [10], result in larger values of the empirical coefficient in the surface renewal model ($A$), with $A$ = 2.43 (Eq (10)) being six times larger than the value reported by [10, 16] ($A$ = 0.42). In a broader evaluation of the coefficient for different wind and density-driven flows, [56] reports values between 0.18 and 1.5. Therefore, based on the above values of $A$, our results appear to be rather high. It is important to consider that despite our robust assessment of the reliability of the range of dissipation rates obtained in this study, there are still large uncertainties associated with estimates of dissipation rates. Assuming an underestimate by a factor of 2 [51] in the dissipation rate by the direct method, we found the coefficient $A$ to become ~1.94. An important factor that may also cause a discrepancy in the value of this coefficient throughout the comparison of different studies is the depth at which the dissipation rate is measured. However, this is not likely to be the reason for the discrepancy between the coefficient of [10, 16] and ours, as the coefficient $A$ = 0.42 was obtained from dissipation measurements made within a few cm of the surface [16] as were we ($z$ = 7.5 cm).

Although the surface renewal model (Eq (11)) is indicated for the water surface, it is difficult to quantify rain-generated dissipation near the air-water interface, where waves, drop-generated bubbles and other processes interplay [11, 16]. According to [11], at the depth of the cavities ($z$ ~ 1 cm), waves may contribute up to 80% of the fluctuating velocity, while the contribution would decrease to less than 20% at $z$ ~ 3 cm. In agreement with those findings, our results showed the best correlation between the dissipation rates and the surface renewal model at 7.5 cm depth instead of closer to the air-water interface.

There are several empirical expressions in the literature for the normalized gas transfer velocity ($k_{600}$) as a function of the rain rate ($R$). Some studies proposed the use of the kinetic energy flux ($F_{KE}$) as the parameter to relate $k_{600}$ with $R$ (Fig 6) [22]. Our results showed that the surface renewal model predicted more of the observed variability in the rain-generated gas transfer velocity than the kinetic energy flux ($r^2$ = 0.87 versus $r^2$ = 0.83), in agreement with [10]. Other studies have proposed direct relationships between $k_{600}$ and $R$. For instance, [23] found a second-order polynomial fitting and [5] suggested a linear relationship based on field data. A parametrization of the laboratory data of [22, 23] has been refitted in [7] yielding a potential relationship where $k_{600} \propto R^{0.704}$, similar to our mechanistic scaling ($k_{600} \propto R^{0.45}$) based on testing 14 rain rates in the wide range between 7 and 89 mm h$^{-1}$.

For practical applications, it is a common assumption that the effects of wind and rain at the interface are linearly independent and additive at low wind speeds [4]. The sum of these contributions can be used as a first-order approximation of the 'total dissipation rate', or it can be used to decide the most suitable scaling relationship [15]. Our scaling expands the available tools, by enabling estimates of turbulence and gas exchange in low-wind environments.

## Broader implications

Improved understanding of the effect of rainfall on gas exchange is crucial for estimating and predicting the fluxes of greenhouse gases, including methane, carbon dioxide, and nitrous oxide, from aquatic ecosystems. The influence of rain on air-water gas exchange is likely most important in inland waters, especially in wind-sheltered systems, where the role of wind in generating near-surface turbulence is limited [22, 23]. Specifically, in tropical regions, rain intensity constitutes a primary factor in seasonal weather changes, and conditions characterized by low wind speed, high rain intensity, and frequent rainfall are commonly found [25]. Therefore, taking into account the impact of rainfall on gas exchange can have major implications for greenhouse gas budgets, not only in terms of short-term dynamics during rain events,

but also for elucidating the seasonal variations in air-water gas exchange in tropical inland waters. Moreover, the exchange of gases in boreal and temperate inland waters [6] as well as in marine systems [24], can be notably influenced by rainfall.

Mechanistic models of greenhouse gas dynamics and fluxes in aquatic environments are typically based on mass balances, where gas transfer velocities are either obtained from measurements or described as a function of mean wind speed [53–55]. The application of the scaling relationships for the gas transfer velocity as a function of the rain rate presented in this study can be used to assess the importance of short-term drivers, such as rain events, on gas dynamics and biogeochemical cycling in marine and inland waters. With readily available data on rainfall as the only additional boundary condition required, future studies can apply the scaling relationships in coupled hydrodynamic and biogeochemical models for a broad range of different aquatic ecosystems and climatic boundary conditions.

In future research, it is advisable to extend investigations to cover experiments that consider the interplay of rain with other turbulence generation mechanisms, such as wind speed and currents. Additionally, researchers are encouraged to conduct more comprehensive characterizations of raindrop size distributions and fall velocities. The validation of our findings through field experiments under real rainy conditions is also recommended.

## Conclusion

For the first time, our experimental results revealed a positive and systematic relationship between rain rate and turbulence at the air-water interface. Rain-induced dissipation rates of turbulent kinetic energy showed a consistent decrease with increasing sampling depth and a logarithmic relationship with rain rate (Eq (12)). We used this empirical model to derive a relationship quantifying the gas transfer velocity as a function of rain rate. In combination, the observed relationships showed good agreement with the surface renewal model (Eq (13)), but with a higher value for the associated empirical constant, than those that have been found in former experiments and for other generation mechanisms of near-surface turbulence, including wind. The findings of this study contribute to the quantitative understanding of the effect of rainfall on gas exchange and can be used to account for rain as a dynamic driver for the fluxes of greenhouse gases from aquatic ecosystems in empirical studies and biogeochemical models.

## Supporting information

**S1 Appendix. Estimate of the accuracy of the PIV system.**
(PDF)

**S1 Table. Comparison of turbulent dissipation rates from PIV_dir (Eq (8)), PIV_spec and ADV (Eq (9)) for all runs at the ADV sampling point location (10 cm depth).**
(PDF)

**S1 Fig.** Pictures of the experimental setup: a) Hose-drying tower (approximately 20 m tall) of the municipal fire brigade in Landau, Germany, in which experiments were conducted. b) Inside view showing the rain generator in the top and the aquarium at the base of the tower. c) detailed view of the rain generator (cf. Fig 1). d) Aquarium with sensors and camera mounting frame.
(PDF)

**S2 Fig. Picture of the rain generator: The outer aluminum frame (1 x 1 x 0.05 m) was filled with water and raindrops formed at regularly arranged holes in the transparent bottom plate.** The rainfall rate was varied by closing or opening a varying number of holes using

adhesive tape (visible as black and silver tape stripes in the picture).
(PDF)

**S3 Fig. Histogram of rain events collected by a meteorological station located on the water surface of Porce III reservoir, Colombia (6˚54'12.6"N, 75˚10'16.1"W).** Data were obtained from measurements with 1h resolution for the periods from 19-Nov-2019 to 09-Mar-2020 and from 01-Sep-2021 to 10-Dec-2021 (n = 81601). **a.** Rainfall rate frequencies, **b.** Rain events frequencies at different wind speed.
(PDF)

**S4 Fig. Time series of pressure measured with the RBR sensor during an experimental series: Without rain and for three different rain rates (see panel headings).** The slopes (in mm min$^{-1}$) and regression statistics (the p-value is the significance of the slope applying the t-test, and $n$ is the number of points used for the fit and t-test) reported in the text boxes are for the linear regressions (red solid lines) using all data (15 min of measurements).
(PDF)

**S5 Fig. Time series of dissolved oxygen concentration at different sampling depth in the aquarium (line color, see legend) observed without rain and for two different rain rates (see panel headings).** Red solid lines show linear regressions that were used for estimating the gas transfer velocity according to Eq (2). The slopes (in μmol L$^{-1}$ min$^{-1}$) in the text boxes are those of linear regressions using all data, the p-value is the significance of the slope applying the t-test, and n is the number of points used for the fit.
(PDF)

**S6 Fig. Example of the analysis results in PIVlab of a pair pre-processed images: Region of interest (dark blue dashed line) extracted from the field of view; fluorescent particles in water (white points); the velocity vectors resulting from the analysis of a pair images (green arrows) and the interrogation areas from the two passes used in the data analysis (zoom at left-bottom in the yellow square).**
(PDF)

**S7 Fig. Spatial distribution of the time-averaged turbulent dissipation rates ($\epsilon_{t\_avg}$) for six different rain rates (see panel titles).**
(PDF)

**S8 Fig. Mean vertical profiles of dissipation rates of turbulent kinetic energy for all measured rain rates (filled symbols).** The solid lines show power-law fits according to the function shown in each legend.
(PDF)

**S9 Fig. Normalized gas transfer velocities $k_{600}$ versus the surface renewal model (Eq (12)) for different water depth at which dissipation rates ($\epsilon$) were measured.** Solid lines show linear regressions according to the equation shown in the legends. The best fit, which was chosen to estimate the empirical coefficient A is highlighted by the green bounding box.
(PDF)

**S10 Fig. Normalized gas transfer velocity $k_{600}$ as a function of the kinetic energy flux of rain reported in Ho et al. (1997) (their Table 1).** The solid line shows a polynomial fit according to the equation shown in the legend.
(PDF)

**S11 Fig. Normalized gas transfer velocity $k_{600}$ as a function of the kinetic energy flux of rain reported in Ho et al. (2000) and Zappa et al. (2009) (taken from Fig 8 in Zappa et al. (2009)).** The solid line shows a polynomial fit according the equation shown in the legend. In the legend, the values in parenthesis report the averaged drop size for freshwater experiments corresponding to the results of Ho et al. (2000) and saltwater experiments correspond to the results of Zappa et al. (2009), in which there was a broad drop size distribution (DSD). (PDF)

## Acknowledgments

Thanks to Christoph Bors and to the team of the Environmental Physics Group of the University of Koblenz-Landau for their valuable support during the experiments. Thanks to the Fire Brigade of Landau, Germany, for allowing us to conduct the experiments at their facilities.

## Author Contributions

**Conceptualization:** Eliana Bohórquez-Bedoya, Lorenzo Rovelli, Andreas Lorke.

**Data curation:** Eliana Bohórquez-Bedoya.

**Formal analysis:** Eliana Bohórquez-Bedoya, Andreas Lorke.

**Funding acquisition:** Eliana Bohórquez-Bedoya, Lorenzo Rovelli, Andreas Lorke.

**Investigation:** Eliana Bohórquez-Bedoya, Lorenzo Rovelli, Andreas Lorke.

**Methodology:** Eliana Bohórquez-Bedoya, Lorenzo Rovelli, Andreas Lorke.

**Project administration:** Andreas Lorke.

**Resources:** Andreas Lorke.

**Supervision:** Andreas Lorke.

**Visualization:** Eliana Bohórquez-Bedoya.

**Writing – original draft:** Eliana Bohórquez-Bedoya.

**Writing – review & editing:** Lorenzo Rovelli, Andreas Lorke.

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
