## [Decision Letter · Decision Letter 0]

2 Jan 2024

PONE-D-23-30722Rainfall as a Driver for Near-Surface Turbulence and Air-Water Gas Exchange in Freshwater Aquatic SystemsPLOS ONE

Dear Dr. Lorke,

Thank you for submitting your manuscript to PLOS ONE. After careful consideration, we feel that it has merit but does not fully meet PLOS ONE’s publication criteria as it currently stands. Therefore, we invite you to submit a revised version of the manuscript that addresses the points raised during the review process.

We look forward to receiving your revised manuscript.

Kind regards,

Erman Ülker

Academic Editor

PLOS ONE

Journal Requirements:

When submitting your revision, we need you to address these additional requirements. 1. Please ensure that your manuscript meets PLOS ONE's style requirements, including those for file naming. The PLOS ONE style templates can be found at  https://journals.plos.org/plosone/s/file?id=wjVg/PLOSOne_formatting_sample_main_body.pdf and https://journals.plos.org/plosone/s/file?id=ba62/PLOSOne_formatting_sample_title_authors_affiliations.pdf 2. Thank you for stating the following financial disclosure:  "This study was partially supported by the German Research Foundation (DFG), project number LO 1150/12-1 (AL) and RO 5921/1-1 (LR). https://www.dfg.de/en/ EBB. was funded by the following programs: -
Research Grants - Short-Term Grants, 2019 (57440917) of the German Academic Exchange Service (DAAD), https://www.daad.co/es-
Scholarship Program No. 757 - National Doctorates of the Ministry of Science, Technology and Innovation of Colombia, https://minciencias.gov.co/-
the Call for Teaching and Student Mobility of 2019-2021 of the Facultad de Minas of the Universidad Nacional de Colombia, " ext-link-type="uri" xlink:type="simple">https://minas.medellin.unal.edu.co/" Please state what role the funders took in the study.  If the funders had no role, please state: "The funders had no role in study design, data collection and analysis, decision to publish, or preparation of the manuscript." If this statement is not correct you must amend it as needed.  Please include this amended Role of Funder statement in your cover letter; we will change the online submission form on your behalf. 3. Please ensure that you refer to Figure 7 in your text as, if accepted, production will need this reference to link the reader to the figure. 4. Please upload a copy of Figures 8 9, to which you refer in your text on pages 24 and 20. If the figure is no longer to be included as part of the submission please remove all reference to it within the text.

Reviewers' comments:

Reviewer's Responses to Questions

**Comments to the Author**

1. Is the manuscript technically sound, and do the data support the conclusions?

Reviewer #1: Yes

Reviewer #2: Yes

2. Has the statistical analysis been performed appropriately and rigorously? 

Reviewer #1: Yes

Reviewer #2: Yes

3. Have the authors made all data underlying the findings in their manuscript fully available?

Reviewer #1: Yes

Reviewer #2: Yes

4. Is the manuscript presented in an intelligible fashion and written in standard English?

Reviewer #1: Yes

Reviewer #2: Yes

5. Review Comments to the Author

Reviewer #1: Comment 1: the abstract part has to introduce the main objective and problem statement for why the study important. It needs modification

Comment 2: on page 4 line number 80 introduction part, you mention overlooking low to moderate of rainfall intensities. Is there any consideration in your study to identify intensity of rainfall?

Comment 3: on page 7 line number 132 you mentioned the inquire additional measurements for rain rates, size and fall velocity of raindrops. Have overcome the significant changes of size and fall velocity in your experiment?

Comment 4: On page 8 line number 159, when you adjusted the number of open holes to target a range of rain rates between 7 and 90 mm h-1, how you identified it’s the most frequent rain rates?

Reviewer #2: Authors are encouraged to engage in a comprehensive discussion of the outcomes, elucidating their potential interpretations in light of prior investigations and the underlying hypotheses. Furthermore, it is essential to deliberate upon the broader implications of the findings and their significance within a wider framework. Additionally, it is advisable to emphasize potential avenues for future research.

6. PLOS authors have the option to publish the peer review history of their article (what does this mean?). If published, this will include your full peer review and any attached files.

Reviewer #1: No

Reviewer #2: **Yes: **Mohamed A. E. AbdelRahman

---

## [Author Response · Author response to Decision Letter 0]

13 Feb 2024

A rebuttal letter that responds to each point raised by the academic editor and reviewer(s). was uploaded as a separate file labeled 'Response to Reviewers'.

---

## [Decision Letter · Decision Letter 1]

20 Feb 2024

Rainfall as a Driver for Near-Surface Turbulence and Air-Water Gas Exchange in Freshwater Aquatic Systems

PONE-D-23-30722R1

Dear Dr. Lorke,

We’re pleased to inform you that your manuscript has been judged scientifically suitable for publication and will be formally accepted for publication once it meets all outstanding technical requirements.

Kind regards,

Erman Ülker

Academic Editor

PLOS ONE

---

## [Editor Report · Acceptance letter]

1 Mar 2024

PONE-D-23-30722R1 

PLOS ONE

Dear Dr. Lorke, 

I'm pleased to inform you that your manuscript has been deemed suitable for publication in PLOS ONE. Congratulations! Your manuscript is now being handed over to our production team.

Kind regards, 

on behalf of

Dr. Erman Ülker 

Academic Editor

PLOS ONE